# Carbon Xerogel Nanostructures with Integrated Bi and Fe Components for Hydrogen Peroxide and Heavy Metal Detection

**DOI:** 10.3390/molecules26010117

**Published:** 2020-12-29

**Authors:** Carmen I. Fort, Mihai M. Rusu, Liviu C. Cotet, Adriana Vulpoi, Ileana Florea, Sandrine Tuseau-Nenez, Monica Baia, Mihaela Baibarac, Lucian Baia

**Affiliations:** 1Department of Chemical Engineering, Faculty of Chemistry and Chemical Engineering, “Babes-Bolyai” University, Arany Janos 11, RO-400028 Cluj-Napoca, Romania; carmen.fort@ubbcluj.ro (C.I.F.); cosmin.cotet@ubbcluj.ro (L.C.C.); 2Laboratory of Advanced Materials and Applied Technologies, Institute for Research-Development-Innovation in Applied Natural Sciences, “Babes-Bolyai” University, Fântânele 30, RO-400294 Cluj-Napoca, Romania; mihaimrusu@gmail.com; 3Department of Condensed Matter Physics and Advanced Technologies, Faculty of Physics, “Babes-Bolyai” University, M. Kogalniceanu 1, RO-400084 Cluj-Napoca, Romania; 4Nanostructured Materials and Bio-Nano-Interfaces Center, Institute of Interdisciplinary Research in Bio-Nano-Sciences, “Babes-Bolyai” University, T. Laurean 42, RO-400271 Cluj-Napoca, Romania; adriana.lazar@ubbcluj.ro; 5LPICM, CNRS, Ecole Polytechnique, IPParis, 91228 Palaiseau, France; lenuta-leana.florea@polytechnique.edu; 6Laboratoire de Physique de la Matière Condensée, Ecole Polytechnique, IPParis, 91228 Palaiseau, France; sandrine.tusseau-nenez@polytechnique.edu; 7Department of Biomolecular Physics, Faculty of Physics, “Babes-Bolyai” University, M. Kogalniceanu 1, RO-400084 Cluj-Napoca, Romania; monica.baia@ubbcluj.ro; 8Laboratory Optical Processes in Nanostructure Materials, National Institute of Materials Physics, Atomistilor str. 405 A, 77125 Bucharest, Romania

**Keywords:** carbon nanocomposites, bismuth, iron, hybrid structures, electrochemical sensors, lead, hydrogen peroxide

## Abstract

Multifunctional Bi- and Fe-modified carbon xerogel composites (CXBiFe), with different Fe concentrations, were obtained by a resorcinol–formaldehyde sol–gel method, followed by drying in ambient conditions and pyrolysis treatment. The morphological and structural characterization performed by X-ray diffraction (XRD), Raman spectroscopy, N_2_ adsorption/desorption porosimetry, scanning electron microscopy (SEM) and scanning/transmission electron microscopy (STEM) analyses, indicates the formation of carbon-based nanocomposites with integrated Bi and Fe oxide nanoparticles. At higher Fe concentrations, Bi-Fe-O interactions lead to the formation of hybrid nanostructures and off-stoichiometric Bi_2_Fe_4_O_9_ mullite-like structures together with an excess of iron oxide nanoparticles. To examine the effect of the Fe content on the electrochemical performance of the CXBiFe composites, the obtained powders were initially dispersed in a chitosan solution and applied on the surface of glassy carbon electrodes. Then, the multifunctional character of the CXBiFe systems is assessed by involving the obtained modified electrodes for the detection of different analytes, such as biomarkers (hydrogen peroxide) and heavy metal ions (i.e., Pb^2+^). The achieved results indicate a drop in the detection limit for H_2_O_2_ as Fe content increases. Even though the current results suggest that the surface modifications of the Bi phase with Fe and O impurities lower Pb^2+^ detection efficiencies, Pb^2+^ sensing well below the admitted concentrations for drinkable water is also noticed.

## 1. Introduction

The detection of water contaminants such as heavy metals represents a significant research focus for present concern due to their significant threat to human health, animal health, and the environment [1]. Additionally, biomolecule detection such as hydrogen peroxide, dopamine, uric acid, etc. plays a crucial role in clinical diagnoses as well as in many industrial applications, such as food processing, pharmaceutical industries, paper bleaching, mineral processing, environmental analysis, and cleaning products [2].

Various analytical methods have been employed for detection of heavy metals or biomarkers including atomic fluorescence spectrometry [3], atomic absorbance spectrometry [4], chemiluminescence [5], chemiresistor [6], electrochemical [7,8,9], and colorimetric [10] sensors. Among most of the above-mentioned methods, which suffer from some technical downsides involving time-consumption, costs, or low sensitivity and selectivity, the electrochemical detection technique shows promising prospects due to its high sensitivity, high selectivity, simple instrumentation, fast response, miniaturization capabilities, portability and low cost [1].

For both biomarker [2] or heavy metal [11] detection, most of sensor preparation methods were based on the immobilization of different type of enzymes (such as horseradish peroxidase, cytochrome, myoglobin, glucose oxidase, acetylcholinesterase, etc.) on the electrodes surface previously functionalized, leading thus to a remarkable selectivity and high sensitivity [2,11]. Such enzyme-based sensors presents notable disadvantages due to the complex enzyme immobilization procedures, the strong influence of the experimental conditions (temperature, humidity, pH, light, etc.) on the sensor stability, and the particular nature of the enzymes, which also greatly influences the sensor stability [1,2,11].

Consequently, many efforts have been dedicated to the development of non-enzymatic electrochemical sensors for heavy metals [1,12] or biomarkers [2] detection. Development of innovative sensitive and cost-effective electrochemical sensors based on different carbonaceous nanocomposite materials, fabricated by different techniques, for the detection of biomolecules, or heavy metal ions has often been in the research focus. Carbonaceous-based nanomaterials such as graphite [13], carbon fibers [14], carbon aerogel [8,15,16], carbon xerogels [7,9], carbon nanotubes (single walled or multi-walled) [17], graphene [18], have been used as the conductive phase in different composite materials suitable for electrochemical detection of biomolecules [2], or heavy metals ions [12]. There is a variety of functional converters/additives such as metal nanoparticles (Au, Ag, Pd, Ir, Sb, St, Hg, and Bi), metal oxides (Bi_2_O_3_, Ce_3_O_4_, Fe_3_O_4_, SnO_2_, NiO, SnO_2_, Co_3_O_4_, MnFe_2_O_4_, MnCo_2_O_4_ NPs, ZnFe_2_O_4_), non-metals (N), halogen (F), alloys (AuPt alloy microsphere) [19,20,21], etc., which can be physically, chemically, or electrochemically introduced into the carbon matrix.

Among this variety of carbonaceous nanocomposite materials, Bi supporting carbon xerogel nanomaterials (CXBi) represent a good candidate for heavy metal detection [7,9], and so far, to the best of our knowledge, the properties of Bi-nanocomposite systems were not fully exploited. As major advantages to the functionality of the binary carbon-metal/oxide system, the CX is endowed with tunable morphological and structural properties and possesses a good electrical conductivity, high porosity and surface area [7,9]. Complementary, Bi adds to the nanocomposite system other important benefits such as (i) good electrochemical properties, i.e., large cathodic potential range and less sensitive to the dissolved oxygen [21]; (ii) its ability to form “fusible alloys” with heavy metals; (iii) good catalytic properties; (iv) low toxicity and environmental friendliness.

In our previous study we reported the use of high mesoporous Fe doped carbon aerogel for modified carbon paste electrodes preparation to catalyze the H_2_O_2_ electroreduction [8]. Based on these findings related to the applications of nanocomposite materials and their electrochemical applications [7,9,22,23], herein we combine the benefits of the presence of both Fe and Bi-based nanoparticles within the carbon xerogel matrix. Thus, this time the aim is to investigate the influence of Fe concentration on both the morphological and structural characteristics as well as the detection efficiency of both heavy metal (Pb^2+^), due to the presence of Bi, and biomarker (H_2_O_2_), due to the presence of Fe.

## 2. Materials and Methods

### 2.1. Reagents

Unless otherwise indicated, reagents were purchased from Sigma Aldrich and were used without any further purification: resorcinol (m-C_6_H_4_(OH)_2_, 99%), formaldehyde solution (37 wt.% in H_2_O, stabilized with methanol, Chem-Lab), bismuth (III) nitrate pentahydrate [Bi(NO_3_)_3_·5H_2_O, 98%, Alfa Aesar], acetic acid (CH_3_COOH, 99.7%), anhydrous iron (II) acetate (Fe(OOCCH_3_)_2_, minimum Fe content 29.5%), acetic acid (CH_3_COOH, 99%), ammonium hydroxide water solution (NH_4_OH, 10 wt.%), glycerol formal (47–67% 5-hydroxy-1,3-dioxane, 33–53% 4-hydroxymethyl-1,3-dioxolane). All reagents were of analytical grade. Bidistilled water was used for the preparation of all solutions.

### 2.2. Synthesis of Bi/Fe/C Xerogel Ternary Composite

The composite synthesis started with the dissolution under stirring of 1.2 g Bi(NO_3_)_3_·5H_2_O in glycerol formal respecting 0.12 g/mL. Then, 2 g resorcinol (R) followed by the formaldehyde (F) solution were added respecting a molar ratio R/F as 0.5. As pH adjustment, 4 mL 10% solution of NH_4_OH was drop by drop added. When 12 mL of acetic acid was poured to the prepared mixture a clear solution was obtained. The iron as Fe(OOCCH_3_)_2_ was dissolute in the solution using different amounts of 0.01, 0.12 and 1.2 g. These are the precursor solution of the final CXBiFe_x_ materials, where x is standing for the initial Fe precursor content added to the synthesis and is 0.01, 0.12 and 1.2 g, respectively. As blank, a solution without iron is also included (CXBiFe_0_).

The obtained solutions were sealed in glass vessels and placed at 60 °C for 3 days. Wet gels with the geometry of the vessels were obtained. These were then rinsed two times with ethanol and hold in acetic acid for one day for washing. After a second step of rinsing with ethanol, the gels were dried in ambient conditions for several days until constant mass. The obtained organic xerogels embedded with Bi and Fe ions were pyrolyzed at 750 °C for 1 h using a heat rate of 3 °C/min and argon atmosphere to yield the final carbon xerogel nanocomposites.

### 2.3. Characterization Methods

X-ray diffraction (XRD) measurements were performed on a powder diffractometer (X’Pert, PANalytical) in Bragg–Brentano geometry, using Cu K_α1_ radiation (graphite monochromator to avoid Fe fluorescence signal) equipped with a Miniprop punctual detector. The experimental setup was as follows: fixed divergence slit 0.5°, fixed anti-scatted slit 1°, fixed incident mask 10 mm, incident, and receiving Soller slits 0.02 rad. The data were collected from 12 to 80° 2θ, step size 0.03°, 20 sec/step. For the phase identification, the ICDD PDF2 (release 2004) was used.

Nitrogen adsorption–desorption analysis was performed with a Sorptomatic (Thermo Electron Corp.) equipment after degassing around 100 mg of the tested material for 20 h at 106 °C in vacuum (<1 mPa). The specific surface area was determined using the three-parameter BET (Brunauer–Emmet–Teller) method, while the pore size distribution and the cumulative pore volume were evaluated using the BJH (Barrett–Joyner–Halenda) model for the mesopore range and the H–K (Horvath–Kawazoe) model for the micropore range.

Raman spectra were recorded with a Renishaw in Via Reflex Raman Microscope equipped with a Ren Cam charge-coupled device (CCD) detector. The 532 nm laser line was used for excitation, and the spectra were collected with a 0.85 NA objective of 100× magnification. Typical integration times were of 30 s, and the laser power was 1 mW. The Raman spectra were recorded with a spectral resolution of 4 cm^−1^.

Scanning electron microscopy (SEM) studies were performed using a FEI Quanta 3D FEG dual beam microscope (FEI, Hillsboro, OR, USA), working in high vacuum mode using ETD (Everhart Thornley Detector). Transmission Electron Microscopy (TEM) analyses have been performed on two different transmission electron microscopes (Jeol 2010, JEOL Ltd., Tokyo, Japan and FEI Titan–Themis, FEI, Hillsboro, OR, USA) both operating at 200 kV accelerating voltage. For the chemical analyses we used a Titan–Themis operating at 200 KV equipped with a Cs probe corrector and a SuperX detector that allows chemical analyses of light and heavy elements through energy dispersive X-ray spectroscopy (EDX) with a spatial resolution within picometer range.

### 2.4. Preparation of the Glassy Carbon/Chitosan (GC/Chi)–CXBiFex Electrodes

Glassy carbon electrode (GCE) surface (with the geometrical area of 0.07 cm^2^) was carefully polished on alumina slurry (1 μm, and then 0.1 μm Stuers, Copenhagen, Denmark). Then, GCE surface was washed with bidistilled water. By sonication for 5 minutes in acetone the alumina particles were removed, concomitantly with other possible contaminants. All CXBiFe_x_ nanocomposites were immobilized onto GCE surface by using a solution of 10 mg chitosan (Chi) polymer in 10 mL of 0.1 M acetic acid. By adding 1 g/L CXBiFe_x_ and sonicated for 2 h, 5 µL of the resulted mixture were placed onto the clean GCE surfaces. Then, by keeping for drying under a beaker for 2 h at room temperature the GC/Chi–CXBiFe_x_ electrodes were obtained.

### 2.5. Electrochemical Measurements

For electrochemical measurements, a PC controlled electrochemical analyzer (AUTOLAB PGSTAT302N EcoChemie, Utrecht, Netherlands) was used. A conventional three-electrode cell, equipped with GC/Chi–CXBiFe_x_ as working electrode, an Ag/AgCl, KCl sat. as reference electrode, and a Pt wire, as counter electrode was used. The electrochemical impedance spectroscopy investigations were carried out at room temperature, by immersing the working electrodes (GC and GC/Chi–CXBiFe_x_), in 0.1 M acetate buffer containing 5 mM [Fe(CN)_6_]^3−^/^4−^, in a frequency range from 104 Hz to 10^−1^ Hz. The electrochemical experiments were performed by cyclic voltammetry (CV) and square wave anodic stripping voltammetry techniques (SWASV). The electroanalytical detection of heavy metal (i.e., Pb^2+^) was carried out in 0.1 M acetate buffer (pH 4.5), after potentiostatic polarization at −1.4 V vs. Ag/AgCl, KCl sat. for 180 s, under constant stirring at 400 rpm. Then, after 10 s of equilibration from the stirring stopping the anodic voltametric scan was achieved. For hydrogen peroxide detection, the SWV investigations were carried out in 0.1 M phosphate buffer (pH 7). The electrochemical behavior of GC/Chi–CXBiFe_x_ was exploited for 1–10 pM Pb^2+^ detection, and 3–30 µM for hydrogen peroxide, respectively. All experiments were carried out at the ambient temperature.

## 3. Results and Discussions

Since a major objective of the present research was to extend the material’s functionality, the synthesis method involved was a sol–gel process based on the polycondensation reaction of resorcinol with formaldehyde that led to the obtaining of ternary composite materials made up from carbon, bismuth and iron components [24]. By adding in the well-adjusted pH reaction medium bismuth and iron salts as metal precursors (i.e., co-synthesis pathway [22]), organic–inorganic wet gels were first achieved. By drying in ambient condition and pyrolysis in inert atmosphere and high temperature (i.e., 750 °C/1 h) ternary xerogels were finally obtained. These are composed of a carbon nanoporous framework embedded with metal/oxide nanoparticles that resulted during the pyrolytic reduction process [22,23,24]. Insights about the structural characteristics that are reflected in their applicability in sensing field are further revealed.

### 3.1. Morphological and Structural Analysis

Due to their wide area of sample analysis and large penetration depth, XRD investigations are initially performed to access the structural information characteristic to the investigated CXBiFe_x_ systems. The acquired diffractograms are presented in Figure 1I. The broad signals centered at 2θ_Cu_ = 25° and 2θ_Cu_ = 44° represent the reflections found in defect rich turbostratic carbon, while the broad signal at about 2θ_Cu_ = 30° can be ascribed to bismuth oxide prior to crystallization. As observed, the crystalline reflections from the tetragonal Bi_2_O_3_ phase (JCPDS file 01-074-1374) are dominant for the CXBiFe_x_ composites with low Fe amounts. With the increase of the Fe concentration, the crystallinity of Bi_2_O_3_ phase is seen to drop, while iron oxide is identified as magnetite Fe_3_O_4_ (JCPDS file 01-075-0449) or maghemite γ-Fe_2_O_3_ (JCPDS file 00-024-0081). Indeed, by XRD, it is difficult to distinguish these two phases as they produce very similar peaks, which can contribute to the broadness of the peaks found at 2θ_Cu_= 35.8°, 57.4° and 63.2° together with the nano-size effects. For the CXBiFe_1.2_ sample, new intense reflections are observed at 2θ_Cu_ = 28.2° and 29.0° corresponding to the mullite phase (Bi_2_Fe_4_O_9_, JCPDS file 00-020-0836).

Raman investigations were performed to detect if the presence of Fe in different concentration induces changes in the graphitization degree of the porous carbon matrix via the catalytic graphitization mechanism observed in previous studies [8]. From the Raman spectra presented in Figure 1II, the D and G signals characteristic to carbon structures are observed for all investigated systems. The first signal around 1348 cm^−1^ is characteristic to A_1g_ defect activated vibrations and the second one around 1595 cm^−1^, due to the in plane E_2g_ phonon found in sp^2^ graphitic carbons [25]. At present conditions, one can observe that the graphitization yield, expressed as I_D_/I_G_ ratio (see Table 1), indicates no clear differences with variation of Fe content. However, as suggested by others, the I_D_/I_G_ is not a fully characteristic figure of merit for carbon structures with small-sized graphene-like basal domains [25]. A certain degree of ordering can still be noticed with the increase of Fe concentration by observing that after a 4-signal deconvolution, the full width at height maxima of D_1_ and G bands (FWHM_D_ and FWHM_G_) decrease with the increase of the Fe concentration (see Table 2). As confirmed by XRD measurements, Fe is essentially found in oxidized state. For this reason, the data suggests that carbothermal reduction reactions and graphitization mechanisms [26] may be active under the given conditions, but only found in an incipient stage.

The effects induced by the variation of Fe concentration over the microporous and mesoporous features of the investigated nanocomposites will affect the associated N_2_ adsorption/desorption isotherms as presented in Figure 1III. The samples with no or intermediate Fe concentrations show type III adsorption isotherms characterized by a convex shaped isotherm with respect to the P/P_0_ axis and no inflexion point at small relative pressure values. This feature is specific to systems with weak adsorbate-adsorbent interactions according to the IUPAC (International Union of Pure and Applied Chemistry) standards [27]. It can be observed that the CXBiFe_1.2_ sample with the highest Fe concentration exhibits a type I isotherm with high N_2_ adsorption amounts and a convex shaped isotherm for P/P_0_ < 0.35 followed by an inflexion point and steady increase until saturation.

The main data derived from the N_2_ isotherms together with the structural parameters achieved from Raman spectroscopy and TEM/SEM/EDX investigations are presented in Table 1 alongside other results already reported for similar C-Bi-based samples. The specific surface area follows a non-monotonous trend: the CXBiFe_0_ sample has the highest value of S_BET_ = 181 m^2^/g and is followed by an abrupt decrease to S_BET_ = 65 m^2^/g for CXBiFe_0.01_, and a further increase until S_BET_ = 162 m^2^/g, for CXBiFe_1.2_ sample. Although the cumulative mesopore volume decreases with the increase of the Fe concentration, the micropore volumes follow an inverted trend. This could be because in some situations, Fe reinforces mesopore walls of the carbon structure. In other cases, by filling of mesopores with Fe nanoparticles micropores could be generated. An optimum ratio between precursors (i.e., C and Bi) permitted to have the highest specific surface area of CXBiFe_0_. The presence of Fe seems to decrease the specific surface area. Also, the higher Fe concentration could increase the loss of Bi content/component during pyrolytic treatment by local increase in temperature. The adsorption data suggest that the synthesis procedure that yielded the highest Fe concentration also tuned the pore formation mechanism towards the micropore range.

Electron microscopy investigations were further required to evaluate the changes in the nanocomposite morphology and structure when the Fe content is increased. As shown in Figure 2, the trapped amounts of Bi and Fe precursors form spheroidal nanoparticle systems imbedded in the pores of the carbon xerogel. Then, as suggested by the topological contrast specific to the SEM micrographs, the nanoparticles are exposed to the surrounding environment during the grinding of the pyrolyzed xerogel monoliths. As presented in Table 1, the elemental composition of the investigated nanocomposites indicates a steady increase of Fe concentration (in at. %) relative to the Bi content, which is kept constant throughout the synthesis step. As seen in the SEM and TEM micrographs and the measured particle size distributions, the average nanoparticle size is between 6-8 nm. At higher concentrations of Fe, the average size of the nanoparticles shifts towards larger values (10 nm), also emphasizing the appearance of secondary mode for nanoparticles with an average diameter of 30 nm for the CXBiFe_1.2_ sample.

Most notably, the high-angle annular dark-field scanning transmission electron microscopy (HAADF)-STEM-EDS analyses demonstrated the nanoparticles found in CXBiFe_1.2_ sample as being hybrid structures, as presented in Figure 3. In the HAADF images, due to the enhanced Z-contrast between Bi and Fe, the free-standing nanoparticles and the regions of the hybrid structures with weaker contrast are associated with Fe-rich phases such as Fe_3_O_4_ or Bi_2_Fe_4_O_9_, while the brighter region are associated with Bi-rich phases such as Bi_2_O_3_. This is well represented in the EDX maps that further confirmed the heterogeneous distribution of Bi, Fe, and O elements within the nanoparticle structure. During pyrolysis at 750 °C, clusters with compositions such as Bi and/or Bi_2_O_3_ and Fe_3_O_4_ are already formed and start to diffuse through the porous xerogel mass.

Having in mind that Bi and Bi_2_O_3_ have smaller bulk melting temperatures (T_Bi_ = 271 °C and T_Bi2O3_ = 817 °C) than Fe and iron oxides (T_Fe_ = 1538 °C and T_Fe2O3-Fe3O4_ ≈ 1567–1597 °C), it is considered that the mobile Bi/Bi_2_O_3_ will migrate more efficiently onto the carbon surface and coalesce with the surrounding clusters. This suggests that the Fe_3_O_4_ clusters will modify the Bi_2_O_3_ nanoparticle growth serving as nucleation and/or anchoring sites. The variations of the morphological and structural features of the nanoparticles strongly depend on the local conditions such as number and size of interacting clusters, local concentration of Bi, Fe and O, and temperature variation with time [28].

As suggested by the electron microscopy and XRD results, at the interface of two or several interacting nanoparticles, new hybrid structures and structural phases such as off-stoichiometric Bi_2_Fe_4_O_9_ could form and to act as anchoring sites for the Bi fraction. This mechanism can also explain the size increase observed at higher Fe concentrations. Thus, as opposed to the more ideal case demonstrated for the CXBiFe_0_ sample, the Bi-based nanoparticles will exhibit a modified surface composition in Fe-integrated systems that may alter their electrochemical response.

### 3.2. Electrochemical Performance

#### 3.2.1. Electrochemical Characterization of GC/Chi–CXBiFe_x_ Electrodes

For characterizing the interfacial properties of GCE/CXBiFe_x_ modified electrodes, the electrochemical impedance spectroscopy (EIS), was used to comparatively investigate the GC/Chi–CXBiFe_x_, and GC electrodes, using [Fe(CN)_6_]^3−^/^4−^ as electrochemical probe (Figure 4). The EIS data were fitted to a modified Randles equivalent circuit [7,9], involving an uncompensated electrolyte solution resistance (Rel) coupled in series with a parallel combination of the interface capacitance (Q) and faradaic impedance. The former symbolizes a mixed capacitance including a constant phase element (CPE) and the double layer capacitance (C), while the faradaic impedance is modelled as a charge transfer resistance (R_ct_) coupled with a mass transfer resistance (W), respectively. By using the ZSimpWin 3.21 software, the values of all above-mentioned parameters were estimated, as can be seen from Table 3.

By comparing the R_ct_ value obtained for bare GC electrode, and the value obtained for all modified electrode, one can conclude that the presence of the high conductive CXBiFe_x_ nanocomposite matrix on the GCE surface led to a significant diminish of the charge transfer resistance (R_ct_). On the other hand, the R_ct_ value increases with the Fe concentration in the CXBiFe_x_ nanocomposite matrix. As previously observed, the nanoparticles are primarily found in oxidized states, also with lower crystallinity and poor conductivity [29]. Such inclusions may also alter the connectivity of the electron conducting matrix, increasing the capacitive response of the system. Moreover, the variation of specific surface area and the micro-porosity values for CXBiFe_x_ nanocomposite matrix, strongly influences the double layer capacitance (C), reflected by following sequence: GC ≪ GC/Chi-CXBiFe_0_ ˂ GC/Chi-CXBiFe_0.01_ ˂ GC/Chi-CXBiFe_0.1_ ˂ GC/Chi-CXBiFe_1.2_.

Interestingly, the following sequence W_CXBiFe0_ ≈ W_CXBiFe0.01_ ≈ W_CXBiFe0.1_ ≈ W_CXBiFe1.2_, showing similar behavior of CXBiFe_x_, was obtained in the region corresponding to lower frequencies, which is the domain attributed to the diffusion limited processes (Figure 4).

#### 3.2.2. Amperometric H_2_O_2_ Detection

The cyclic voltammograms, recorded at GC/Chi-CXBiFe_1.2_ nanocomposite modified electrode (Figure 5), in the absence and in the presence of 1 mM H_2_O_2_, showed an electrocatalytic activity toward the H_2_O_2_ reduction at GC/Chi-CXBiFe_1.2_. The presence of 1 mM H_2_O_2_ in the electrolyte solution, in the potential domain, which corresponds to the voltametric peak due to the existence of Fe oxides in CXBiFe1.2 matrix, lead to a significant increase of the reduction peak current, thus demonstrating an electrocatalytic process. Therefore, two consecutive steps are involved: (i) the first one, occurring according to a Fenton-type mechanism [30], is the catalytic oxidation of reduced iron states ions by H_2_O_2_; (ii) the second one, assuring the regeneration of the catalyst, is the electrochemical reduction of the chemically generated Fe^3+^ ions.

The electroanalytical parameters for H_2_O_2_ reduction were estimated from the amperometric calibration curves, recorded at the GC/Chi–CXBiFe_x_ (Figure 6). After successive injections of 3 µM H_2_O_2_ typical current time response curves were obtained for all four investigated electrodes (Figure 6A). The amperometric response provided by the GC/Chi–CXBiFe_x_ becomes stable in less than 6 s, making these electrodes competitive with other similar sensors [8]. The average results, obtained by using three different GC/Chi–CXBiFe_x_ modified electrodes and electrolyte solutions containing H_2_O_2_ (1–30 µM) were used to draw the calibration curves described in Figure 6B. The corresponding linear regressions parameters illustrated in Table 4 enable the calculation of the electroanalytical parameters that are further compared with previously reported results for H_2_O_2_ detection (Table 5).

The increase in sensitivity values, and the decrease in detection limit values (estimated for a signal to noise ratio of 3) for H_2_O_2_ at GC/Chi–CXBiFe_x_ modified electrodes, with the Fe concentration increasing in CXBiFe_x_ nanocomposite (from 0 to 1.2) was observed (Table 5). As expected, the presence of nanostructured iron oxides in the CXBiFe_x_ nanocomposite matrix, successfully led to improved electroanalytical parameters values with the increase in the amount of iron precursor. The obtained electroanalytical parameters, recommend the GC/Chi–CXBiFe_x_ modified electrodes as competitive for H_2_O_2_ detection, with better or comparable results with others already published (Table 5).

GC, glassy carbon; Chi, chitosan; CX, carbon xerogel; CA, carbon aerogel; AP-Ni-MOF, Ni^2+^ metal organic framework based on adipic acid piperazine; CoFe/NGR, CoFe nanoparticles on the nitrogen-doped graphene; PFECS, polymer (poly(2,5-bis((2-ferroceneylethyl)oxy carbonyl)styrene).

#### 3.2.3. SWASV for Pb^2+^ Detection

The recorded voltammograms for GC/Chi–CXBiFex modified electrodes show well-shaped anodic peaks, corresponding to the dissolution of Pb previously deposited on the electrodes surface during the preconcentration step (Figure 7A,C).

By using three different GC/Chi–CXBiFe_x_ modified electrodes, for each Fe concentration, and electrolyte solutions containing very low Pb^2+^ concentrations (1–10 pM) the SWASV were recorded. Thus, the average results were used to draw the calibration curves for each type of electrode (Figure 7B,D). The obtained linear regression parameters (Table 6) permit the calculation of the electroanalytical parameters for Pb^2+^ detection (Table 7).

The anodic peak potential values for Pb^2+^ detection at the prepared GC/Chi–CXBiFe_x_ modified electrodes present small differences (Table 6) that can be associated with the material surface hydrophobicity [7].

Moreover, by the increasing of the Fe concentration in CXBiFe_x_ nanocomposite from 0 to 1.2%, the decreasing of the sensitivity from 1.17·10^6^ µA/µM to 6.39·10^5^ µA/µM and the increasing of the detection limit values from 0.36 pM to 1.24 pM (estimated for a signal to noise ratio of 3) was observed (Table 6). A possible explanation for the variation of the obtained electroanalytical parameters can arise from the corroborated effect of the (i) partial coverage of the Pb^2+^ sensing Bi centers due to Bi-O-Fe interactions and the formation of hybrid nanostructures (ii) individual or cumulative effects of the size and spatial distribution of the Bi/Fe nanoparticles, and (iii) charge transport properties affected by matrix graphitization yields, the chemical state of the nano-inclusions and the porosity (Table 1). Still, excellent electroanalytical performance (sensitivity, detection limit and linear range) was obtained at the GC/Chi–CXBiFe_x_, which can detect Pb^2+^ concentrations starting from much lower values than the ones reported by official safety and recommendation standards for drinkable water [36,37]. This is mainly due to the irregular microstructure of CXBiFe_x_ nanocomposite, where the Bi/Fe nanoparticles, randomly dispersed in the carbon xerogel, offer an easy access to the heavy metal ions. The obtained analytical parameters, sensitivity, and detection limit, recommend the GC/Chi–CXBiFe_x_ modified electrode as competitive for Pb^2+^ detection, with comparable results with the best already published (Table 7).

By comparing four different compositions of Fe-modified carbonaceous nanocomposite materials, CXBiFe_x_, the good electroanalytical properties, for both heavy metals (Pb^2+^), and biomarkers (H_2_O_2_) detection, proved that the synthetized electrode materials are well-matched with the two different applications. Due to Bi-O-Fe interactions, the H_2_O_2_ and Pb^2+^ sensing performances do not evolve in tandem: as Fe is seen to hinder the sensing capabilities of Bi while improving the H_2_O_2_ detection through its inherent Fenton mechanisms. The present study proved that CXBiFe_x_ composite material open new opportunities for sensors development, offering the advantages of using a very low amount of nanomaterial (CXBiFe_x_) for the electrode preparation, and bifunctionalities.

## 4. Conclusions

Xerogel nanocomposites were obtained by tailoring the initial resorcinol–formaldehyde synthesis with Bi and Fe precursors. During this study, the variation of Fe concentration is investigated, while keeping constant the other synthesis parameters, including the Bi concentration. The increased Fe content, combined with the pyrolysis effect induced significant changes at the nanoscale. First, the growth of β-Bi_2_O_3_ and amorphous Bi_2_O_3_ was altered due to Bi-O-Fe interactions, which ultimately led to hybrid nanoparticles with increased size and structural features resembling that of a defected Bi_2_Fe_4_O_9_ structure. Secondly, the excess Fe amounts will be introduced as Fe_3_O_4_ nanoparticles as observed in XRD and TEM results. Finally, considering the changes in features of the carbon support, higher graphitization yields were not detected with the increase in Fe content, as pure metal/carbide catalytic states were not reached during present thermal conditions. Nevertheless, changes of the porous features for the carbon support were induced as the specific surface area and micropore volumes increased with Fe concentration. The electroanalytical parameters values indicate that the presence of Fe in the CXBiFe_x_ nanocomposite decrease the Pb^2+^ detection efficiency, most probably due to modification of Bi nanoparticle surface with Fe phase. Nevertheless, the obtained composites where still operational for sensing Pb^2+^ concentrations well below the standard detection limits. Further on, the nanocomposites revealed improved performance for H_2_O_2_ detection with the increase of Fe content. This clearly indicates that such materials are compatible with the two different applications and may represent a starting point for contexts where heavy metal ions and biological environments interact.

## Figures and Tables

**Figure 1 molecules-26-00117-f001:**
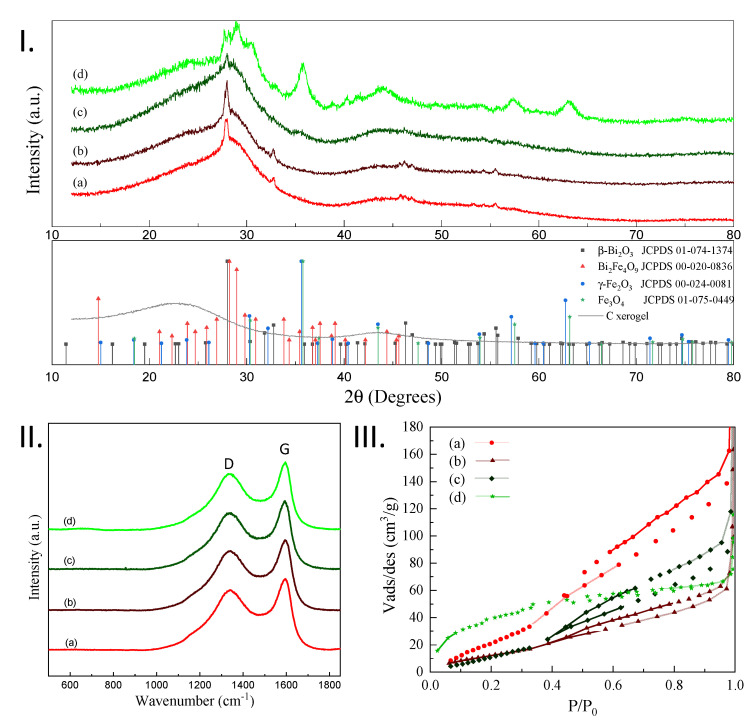
Characterization of (a) CXBiFe_0_, (b) CXBiFe_0.01_, (c) CXBiFe_0.12_ and (d) CXBiFe_1.2_ investigated samples through (**I**). XRD with attached reference signals specific to bare carbon xerogel, β-Bi_2_O_3_, mullite Bi_2_Fe_4_O_9_, maghemite Fe_2_O_3_, and Fe_3_O_4_ magnetite (**II**). Raman spectroscopy revealing the D and G carbon specific vibrational region, and (**III**). N_2_ adsorption/desorption isotherms.

**Figure 2 molecules-26-00117-f002:**
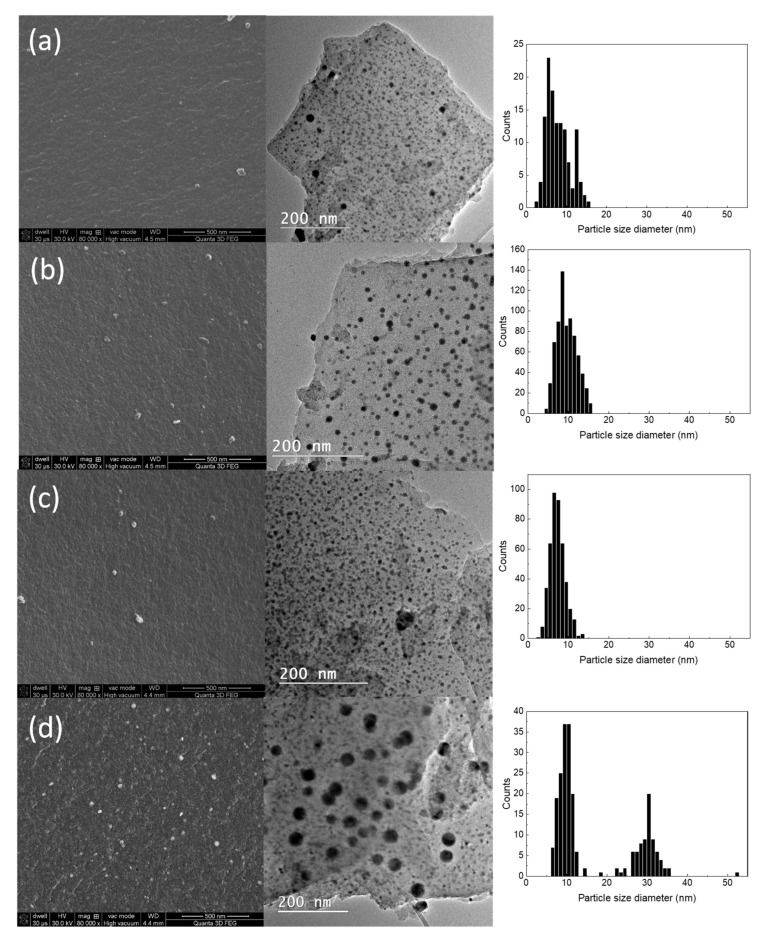
SEM and TEM images together with nanoparticle size histograms corresponding to samples (**a**) CXBiFe_0_, (**b**) CXBiFe_0.01_, (**c**) CXBiFe_0.12_ and (**d**) CXBiFe_1.2_.

**Figure 3 molecules-26-00117-f003:**
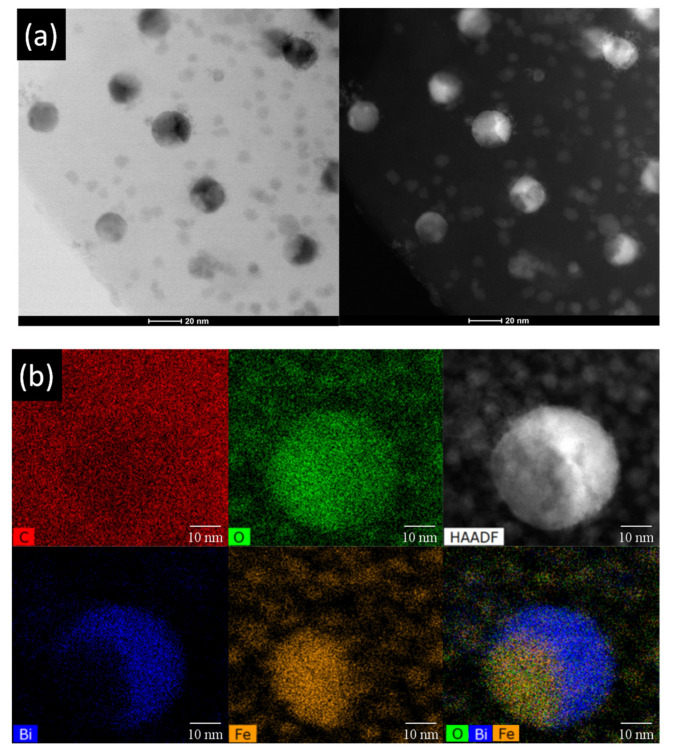
(**a**) BF-STEM and HAADF-STEM images on a large area of the sample containing the hybrid nanoparticles; (**b**) STEM-EDS elemental map and its corresponding HAADF image on a single hybrid nanoparticle showing its chemical composition with bismuth in dark blue, iron in orange, and oxygen in green demonstrated in sample CXBiFe_1.2_.

**Figure 4 molecules-26-00117-f004:**
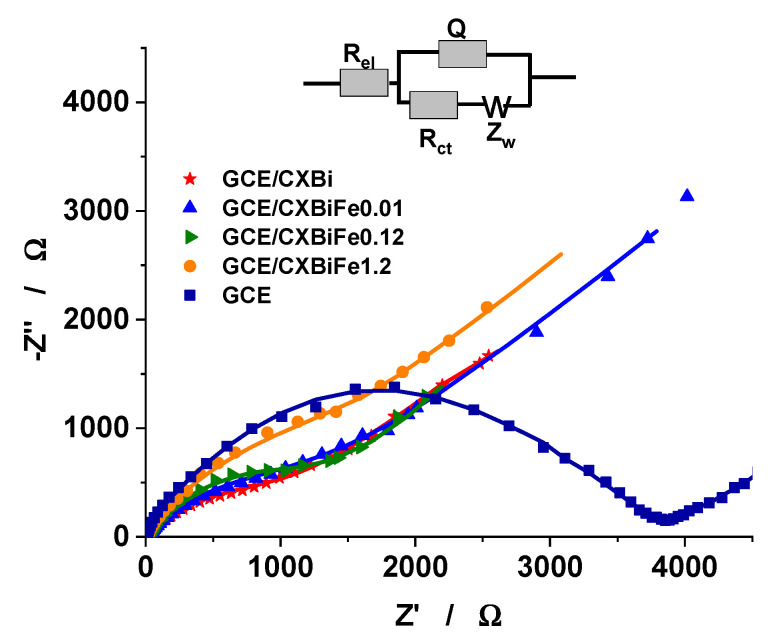
EIS spectra of GCE and GCE/Chi–CXBiFe_x_. Experimental conditions: supporting electrolyte, 0.1 M acetate buffer (pH 4.5) containing 1 mM [Fe(CN)6]^3−/4−^; applied potential, 0.2 V vs. Ag/AgCl, KClsat, frequency interval, 0.1–10^4^ Hz.

**Figure 5 molecules-26-00117-f005:**
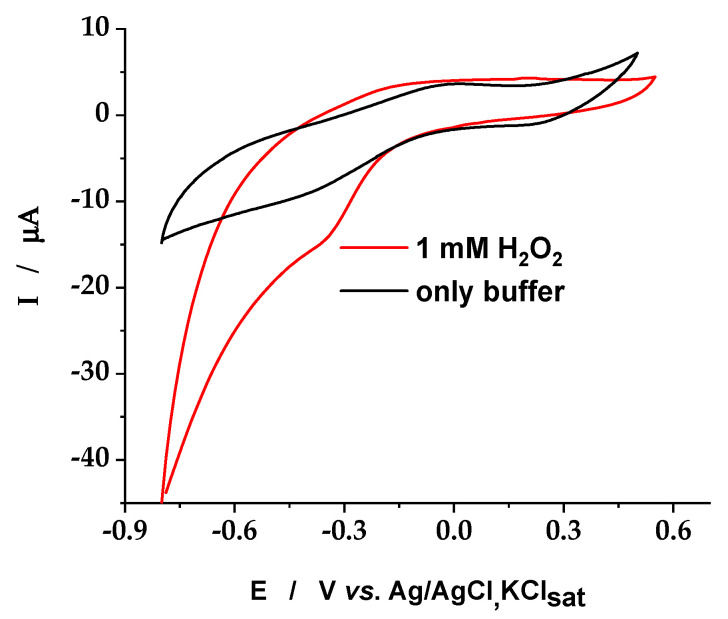
Cyclic voltammograms recorded in the absence and the presence of 1 mM H_2_O_2_ at GC/Chi-CXBiFe_1.2_. Experimental conditions: scan rate, 20 mV s^−1^; supporting electrolyte, 0.2 M phosphate buffer (pH 7); starting potential, −0.8 V vs. Ag/AgCl, KCl sat.

**Figure 6 molecules-26-00117-f006:**
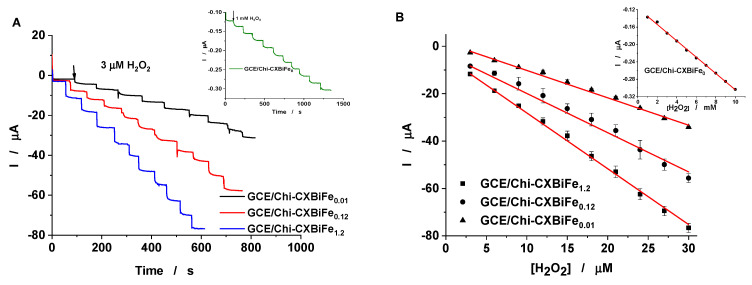
I vs. time dependence recorded at GC/Chi–CXBiFe_x_, for successive addition of 3 µM H_2_O_2_, and 1 mM H_2_O_2_, respectively (**A**), and the corresponding amperometric calibration curve (**B**). Experimental conditions: rotating speed 400 rpm; supporting electrolyte, 0.2 phosphate buffer M (pH 7); applied potential, −0.3 V vs. Ag/AgCl, KCl sat.

**Figure 7 molecules-26-00117-f007:**
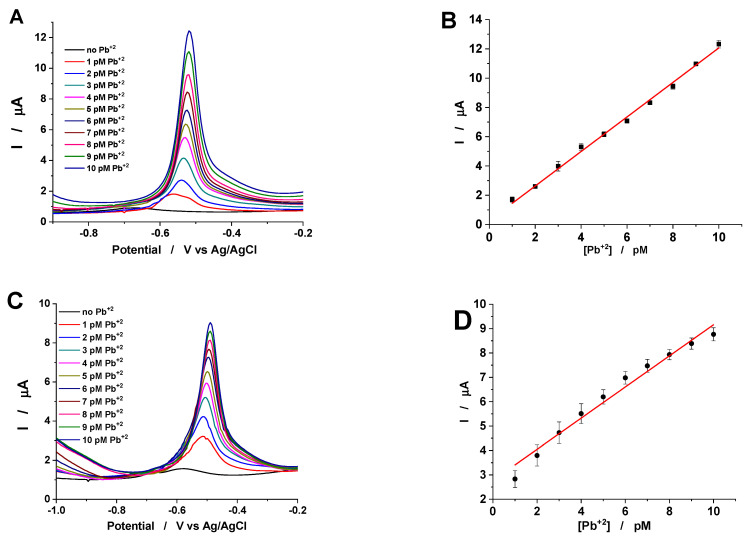
SWASVs recorded at GC/Chi-CXBiFe_0_ (**A**) and GC/Chi-CXBiFe_1.2_ (**B**) electrodes in the presence and absence of Pb^2+^, and the corresponding calibration curve (**C**,**D**), respectively. Experimental conditions: supporting electrolyte, 0.1 M acetate buffer (pH 4.5); deposition potential, −1.4 V vs. Ag/AgCl, KCl sat; deposition time, 180 s; frequency, 10 Hz; amplitude, 25 mV; starting dissolution potential, −1.2 V vs. Ag/AgCl, KCl sat.

**Table 1 molecules-26-00117-t001:** Summary of morphological and structural parameters obtained for the synthesized samples and other similar materials.

Material	Preparation Method	I_D_/I_G_	S_BET_ (m^2^/g)	V_mpores_ (cm^3^/g)	V_μpore_ (cm^3^/g)	<D> (nm)	C:O:Fe:Bi (at%)	Ref.
C-Bi xerogels	impregnation	-	80	0.041	-	56	1% (Bi)	[9]
		-	200		-	-	7% (Bi)	[22]
	co-synthesis	-	-	-	-	85/155	16.4% (Bi)	
		-	400	-	-	29/90	4.08% (Bi)	
		-	71.5	0.049	-	100	6% (Bi)	
		0.90	142	0.077	-	25/110	95.9:3.2:0:0.9	[7]
C-Bi aerogels	co-synthesis	0.89	570	1.012		60/120	96.8:2.7:0:0.5	[7]
CXFe	co-synthesis	0.85	344	-	-	18	94.2:5.2:0.2:0.4	[23]
CXBiFe_0_	co-synthesis	0.84	185	0.250	0.050	8	94.2:2.1:0.0:0.2	Present work
CXBiFe_0.01_	co-synthesis	0.84	65	0.093	0.025	10	96.3:3.5:0.02:0.1	
CXBiFe_0.12_	co-synthesis	0.82	79	0.161	0.026	7	96.6:3.0:0.2:0.1	
CXBiFe_1.2_	co-synthesis	0.82	162	0.038	0.071	10/33	96.4:2.9:0.6:0.03	

**Table 2 molecules-26-00117-t002:** Results obtained after a four peaks deconvolution of the D–G region of the Raman spectra.

Sample	Peak Index	Position (cm^−1^)	FWHM (cm^−1^)	Peak Area (%)
CXBiFe_0_	D_4_	1244.85	232.98	24.37
	D_1_	1348.35	142.76	29.00
	D_3_	1530.94	194.04	33.63
	G	1595.87	63.46	13.00
CXBiFe_0.01_	D_4_	1241.63	231.16	22.64
	D_1_	1347.84	145.10	31.41
	D_3_	1533.29	189.65	32.91
	G	1597.13	62.57	13.03
CXBiFe_0.12_	D_4_	1249.00	244.48	25.63
	D_1_	1346.06	139.15	28.47
	D_3_	1534.51	196.54	33.26
	G	1594.07	59.55	12.63
CXBiFe_1.2_	D_4_	1249.00	253.17	25.28
	D_1_	1344.53	137.20	29.13
	D_3_	1536.29	199.32	32.70
	G	1595.01	58.03	12.88

**Table 3 molecules-26-00117-t003:** The parameters of the equivalent circuit.

	Electrode				
	GC	GC/Chi-CXBiFe_0_	GC/Chi-CXBiFe_0.01_	GC/Chi-CXBiFe_0.1_	GC/Chi-CXBiFe_1.2_
R_el_ (Ω/cm^2^)	22.23 ± 2.30	48.19 ± 0.56	23.72 ± 5.28	23.32 ± 5.07	13.92 ± 5.74
CPE_dl_ (µS·s^n^/cm^2^)	0.60 ± 3.70	65.1 ± 1.58	77.4 ± 3.93	70.88 ± 1.13	381 ± 14.12
n	0.78 ± 0.83	0.57 ± 1.35	0.65 ± 0.83	0.68 ± 2.58	0.72 ± 3.98
R_ct_ (Ω/cm^2^)	3904 ± 3.75	1247 ± 1.43	1801 ± 3.75	2187 ± 6.20	3654 ± 5.04
W (mS·s^1/2^/cm^2^)	0.66 ± 5.68	0.49 ± 1.70	0.57 ± 3.22	0.54 ± 1.00	0.60 ± 3.99
C (µF/cm^2^)	0.10	9.79	27.13	29.48	433.30
χ^2^	0.003552	0.0003785	0.00140	0.00286	0.00514

n is the roughness factor; (±) represents the relative standard deviation (%).

**Table 4 molecules-26-00117-t004:** Linear regression parameters for amperometric detection of H_2_O_2_ at GC/Chi–CXBiFe_x_.

Electrode Type	Intercept (µA)	Slope (µA/µM)	R^2^	N
GC/Chi-CXBiFe_1.2_	−4.586 ± 0.185	−2.354 ± 0.028	0.99864	10
GC/Chi-CXBiFe_0.12_	−3.353 ± 0.235	−1.655 ± 0.053	0.99074	10
GC/Chi-CXBiFe_0.01_	1.365 ± 0.327	−1.155 ± 0.021	0.99679	10
GC/Chi-CXBiFe_0_	−0.116 ± 0.001	−1.879 × 10^−5^ ± 0.028 × 10^−5^	0.99796	10

**Table 5 molecules-26-00117-t005:** Analytical parameters of the sensors based on carbonaceous materials used for amperometric detection of H_2_O_2_.

Electrode Type	Applied Potential V vs. Ag|AgCl, KCl_sat_	Linear Range	Detection Limit (µM)	Ref.
(Fe-CA)-CPE	−0.3	1–50 mM	500	[8]
GC/Chi-BiFeCX	−0.3	5–50 µM	4.77	[23]
GC/Chi-BiFeCX-TiO_2_	−0.3	5–80 mM	3110	[23]
GCE/RGO/Au/Fe_3_O_4_/Ag	0.55	2 µM–1.2 mM	1.43	[31]
PFECS/rGO/GCE	0.44	10–190µM	1.25	[32]
CoFe/NGR	−0.25	1–8654	0.28	[33]
AP-Ni-MOF/CPE	−0.25	4 µM–60 mM	0.9	[34]
MnO_2nanosheets_/GCE	−0.6	25 nM~2 μM and 10~454 μM	5 nM	[35]
GC/Chi-CXBiFe_0_	−0.3	1–10 mM	842.24	This work
GC/Chi-CXBiFe_0.01_	−0.3	3–30 µM	0.85	
GC/Chi/CXBiFe_0.12_	−0.3	3–30 µM	0.43	
GC/Chi-CXBiFe_1.2_	−0.3	3–30 µM	0.24	

**Table 6 molecules-26-00117-t006:** Linear regression parameters for SWASV detection of Pb^2+^ at CXBiFe_x_ nanocomposites modified glassy carbon electrodes.

Electrode Type	Intercept (µA)	Slope (µA/pM)	R^2^	N
GC/Chi-CXBiFe_0_	2.69 ± 0.14	1.17·10^6^ ± 0.02·10^6^	0.99747	10
GC/Chi-CXBiFe_0.01_	2.98 ± 0.18	1.01·10^6^ ± 0.02·10^6^	0.99366	10
GC/Chi/CXBiFe_0.12_	1.20 ± 0.09	3.77·10^5^ ± 0.16·10^5^	0.99001	7
GC/Chi-CXBiFe_1.2_	2.76 ± 0.26	6.39·10^5^ ± 0.37·10^5^	0.97323	10

**Table 7 molecules-26-00117-t007:** Analytical parameters of the sensors based on carbonaceous materials used for SWASV detection of Pb^2+^.

Electrode Type	Peak Potential V vs. Ag|AgCl, KCl sat.	Linear Range	Sensitivity (µA/µM)	Detection Limit (pM)	Ref.
GC/Chi-(Bi-CX)	−0.55	1–10 pM	1.15·106	0.36	[9]
GC/Chi-(Bi-CX)	−0.56	1–10 pM	1.37·106	0.28	[7]
GC/Chi-(Bi-CA)	−0.44	1–10 pM	2.3·105	0.48	[7]
GC/Chi-CXBiFe_0_	−0.53	1–10 pM	1.17·106	0.36	This work
GC/Chi-CXBiFe_0.01_	−0.58	1–10 pM	1.01·106	0.54	
GC/Chi-CXBiFe_0.12_	−0.56	1–10 pM	3.77·105	0.77	
GC/Chi-CXBiFe_1.2_	−0.51	1–10 pM	6.39·105	1.24	

GC, glassy carbon; Chi, chitosan; CX, carbon xerogel; CA, carbon aerogel.

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
