# Peer review of "Carbon Xerogel Nanostructures with Integrated Bi and Fe Components for Hydrogen Peroxide and Heavy Metal Detection"

_molecules, 2020, doi:10.3390/molecules26010117_

Round 1
Reviewer 1 Report
These comments are in regards to the manuscript entitled “Carbon xerogel nanostructures with integrated Bi and Fe components for hydrogen peroxide and heavy metals detection” by C. I. Fort and co-workers. In this work, the authors employ a resorcinol/formaldehyde synthesis followed by pyrolysis to obtain carbon xerogels functionalized with bismuth and various concentrations of iron. These xerogels are dispersed into chitosan solutions and then deposited onto glassy carbon substrates for use as novel nanocomposite electrodes. The premise for adding bismuth is to aid in the detection of heavy metal contaminants in water supplies, specifically for lead in this study. The premise for adding iron is to catalyze the reduction of hydrogen peroxide. In this work, the authors state their goal of developing a multifunctional electrode, presumably one that can perform both assays at high sensitivity. In the ideal case where the two metals have no interaction with each other, this result would be expected. However, in the manuscript it is shown that the inclusion of iron has a very large impact on the properties of the resulting nanocomposite electrode, especially as revealed by impedance spectroscopy. Interesting morphological data suggest that the iron may act as a nucleation center for the more mobile bismuth during pyrolysis, resulting in porosity changes and in the growth of nanoparticles with size dependent on the initial iron load. At the highest iron concentrations there is evidence of the formation of new hybrid bismuth/iron oxide nanostructures. Any of these structural changes could reasonably be expected to affect the electrochemical response of the material.
Unsurprisingly, the electrodes with the least amount of iron (preferably none) are most sensitive for lead detection, while the electrodes with the most iron perform the best hydrogen peroxide reduction. What is interesting is that the combination of Bi and Fe in this study seems to perform hydrogen peroxide reduction better than Fe alone from previous studies (reference 8, also by the author), although a direct comparison is difficult as the electrode and the iron concentrations in the two studies are very different. Table 1 seems to suggest that the authors may have previously fabricated a carbon xerogel naocomposite electrode with Fe only (no Bi), but I can’t find Ref 17 anywhere online. Perhaps it was cited incorrectly? At any rate, it would be interesting to know what role, if any, that the added Bi plays in peroxide reduction using this type of electrode, or if even better results could be obtained by completely eliminating the Bi component. Perhaps this could be addressed in a future study. If it has been studied previously, some additional discussion in the manuscript would make a nice addition.
The novel electrodes presented here are thoroughly characterized and demonstrate very good detection limits, better than most comparable studies in the literature. I recommend that this manuscript be published with minor revisions as outlined below. However, I strongly suggest that the manuscript would benefit from editing by a native English speaker, if the Journal provides this service. Moreover, while reviewing this manuscript, a few of the cited references that I pulled up don’t seem to match their use in the text. The ones I found are noted below. I suspect there are more. The entire list should be carefully checked and corrected.
(1) On line 115, I’m assuming the author meant to list the first value of Fe as 0.01g and not 0.1 g?
(2) In Table 3, what is the parameter "n"?
(3) In Table 3, the error bars on the Rel value for the electrode with no iron seem surprisingly low. Did the author mean for them to be +/- 5.6 instead of +/- 0.56?
(4) Reference 17 does not seem to exist. Perhaps this has been cited incorrectly?
(5) In Table 7, entries 2 and 3 don’t seem to correspond to Reference 7. The cited study is for conducting polymer (polyaniline) nanowires, not carbon xerogels. Moreover, there are no results for lead detection in this paper.
(6) References 6 and 7 are identical.
(7) In Table 3, Reference 27 doesn’t seem to correspond to the entry in the table. They did build an H2O2 sensor, but they used RGO/Fe3O4 and reported limits of detection of 3.2 µM, not 0.28 µM.
Author Response
These comments are in regards to the manuscript entitled “Carbon xerogel nanostructures with integrated Bi and Fe components for hydrogen peroxide and heavy metals detection” by C. I. Fort and co-workers. In this work, the authors employ a resorcinol/formaldehyde synthesis followed by pyrolysis to obtain carbon xerogels functionalized with bismuth and various concentrations of iron. These xerogels are dispersed into chitosan solutions and then deposited onto glassy carbon substrates for use as novel nanocomposite electrodes. The premise for adding bismuth is to aid in the detection of heavy metal contaminants in water supplies, specifically for lead in this study. The premise for adding iron is to catalyze the reduction of hydrogen peroxide. In this work, the authors state their goal of developing a multifunctional electrode, presumably one that can perform both assays at high sensitivity. In the ideal case where the two metals have no interaction with each other, this result would be expected. However, in the manuscript it is shown that the inclusion of iron has a very large impact on the properties of the resulting nanocomposite electrode, especially as revealed by impedance spectroscopy. Interesting morphological data suggest that the iron may act as a nucleation center for the more mobile bismuth during pyrolysis, resulting in porosity changes and in the growth of nanoparticles with size dependent on the initial iron load. At the highest iron concentrations there is evidence of the formation of new hybrid bismuth/iron oxide nanostructures. Any of these structural changes could reasonably be expected to affect the electrochemical response of the material.
The referee is right. At the highest iron concentrations, the newly formed hybrid bismuth/iron oxide nanostructures influence the electrochemical response of the material. Thus, the increasing of the detection limit and the decreasing of the sensibility for Pb2+ detection was observed. This result proved that the Bi particles responsible forPb2+ detection are covered by iron oxide. Moreover, a decreasing of detection limit and an increasing of sensitivity for H2O2 detection was obtained.
Unsurprisingly, the electrodes with the least amount of iron (preferably none) are most sensitive for lead detection, while the electrodes with the most iron perform the best hydrogen peroxide reduction. What is interesting is that the combination of Bi and Fe in this study seems to perform hydrogen peroxide reduction better than Fe alone from previous studies (reference 8, also by the author), although a direct comparison is difficult as the electrode and the iron concentrations in the two studies are very different.
The reviewer observation is correct. A direct comparison with our previous results from ref. 8 is impossible to be done because there are many differences (i.e., structural environment, but especially morphological particularities) between those works:
- the electrodes are completely different. Now a GCE was used in order to be modified, and in our previous work was used a carbon paste electrode (prepared from graphite powder, carbon aerogel modified with iron oxide, and paraffin oil);
- the carbon matrix used for iron/bismuth modification is different. In the previous work was used carbon aerogel, and in this work the matrix was carbon xerogels;
- the iron concentrations in the two studies are very different;
- the Bi is present in this study, and is missing in the previous one, but the porous networks are different from the structural and morphological perspective.
A possible explanation for the better results from this work in comparison with the previous one can be the better exposure of the nanocomposite electrode material to the electrolyte solution. Thus, even a very low quantity of material was used for electrode preparation, the active surface is higher in this case. Comparatively, in the carbon paste electrode, the active surface (carbon aerogel modified with iron oxides) is covered by graphite powder and paraffin oil, thus a low active surface is in contact with the electrolyte solution.
Table 1 seems to suggest that the authors may have previously fabricated a carbon xerogel nanocomposite electrode with Fe only (no Bi), but I can’t find Ref 17 anywhere online. Perhaps it was cited incorrectly?
The reviewer is right; the reference was cited incorrectly. In the revised manuscript the indicated reference together with all other references were carefully checked and the errors were corrected.
At any rate, it would be interesting to know what role, if any, that the added Bi plays in peroxide reduction using this type of electrode, or if even better results could be obtained by completely eliminating the Bi component. Perhaps this could be addressed in a future study. If it has been studied previously, some additional discussion in the manuscript would make a nice addition.
The reviewer pointed out an interesting idea. Until now only Bi component in carbon xerogels matrix was used to prepare GCE modified for water peroxide electrocatalytic reduction (Figure 6). For this modified electrode, low electrocatalytic current for H2O2 was recorded in comparison with the electrodes prepared with Fe-composite materials. Taking into account the synthesis procedure of the Bi and Fe carbon xerogel nanocomposite materials (by sol-gel method), the absence of Bi precursor in the synthesis procedure, lead to a matrix of the final composite which is not the same, neither structurally nor morphologically. On the other hand, the aim of our study was to obtain a sensitive electrode material for both, heavy metals and H2O2 detection. A composite electrode material based only on iron oxides can be indeed the subject of another works.
The novel electrodes presented here are thoroughly characterized and demonstrate very good detection limits, better than most comparable studies in the literature. I recommend that this manuscript be published with minor revisions as outlined below. However, I strongly suggest that the manuscript would benefit from editing by a native English speaker, if the Journal provides this service.
The whole manuscript was double checked aiming to eliminate the spelling errors and, at the same time, to improve the English level.
Moreover, while reviewing this manuscript, a few of the cited references that I pulled up don’t seem to match their use in the text. The ones I found are noted below. I suspect there are more. The entire list should be carefully checked and corrected.
(1) On line 115, I’m assuming the author meant to list the first value of Fe as 0.01g and not 0.1 g?
As mentioned above all references were carefully checked and the errors were corrected. Additionally, the phrase mentioned by the reviewer, which is related to the Fe amount, was corrected accordingly (line 117).
(2) In Table 3, what is the parameter "n"?
The parameter “n” in Table 3 represents the roughness factor, which has the value between 0 and 1. The parameter was defined in the text on line 334.
(3) In Table 3, the error bars on the Rel value for the electrode with no iron seem surprisingly low. Did the author mean for them to be +/- 5.6 instead of +/- 0.56?
We carefully check the parameters and the errors values from Table 3, and the error value of +/- 0.56% is correct.
(4) Reference 17 does not seem to exist. Perhaps this has been cited incorrectly?
The referee was right, we completed the reference 17 with doi 10.1166/jnn.2020.18963 as the article is accepted, but at this moment the paper has no other identification information (Fort, C.I., Rusu, M.M.; Pop, L.C.; Cotet, L.C.; Vulpoi, A.; Baia, M.; Baia, L. Preparation and Characterization of Carbon Xerogel Based Composites for Electrochemical Sensing and Photocatalytic Degradation. J. Nanosci. Nanotechnol., accepted, doi 10.1166/jnn.2020.18963.).
(5) In Table 7, entries 2 and 3 don’t seem to correspond to Reference 7. The cited study is for conducting polymer (polyaniline) nanowires, not carbon xerogels. Moreover, there are no results for lead detection in this paper.
The reviewer is right. The references were carefully checked and the mistake was corrected (line 478).
(6) References 6 and 7 are identical.
The reference was replaced with the corrected one (line 478).
(7) In Table 3, Reference 27 doesn’t seem to correspond to the entry in the table. They did build an H2O2 sensor, but they used RGO/Fe3O4 and reported limits of detection of 3.2 µM, not 0.28 µM.
The reviewer is right. The reference was carefully checked and the error was corrected. The Reference 27 cited in Table 5 in the initially submitted paper appears in the revised manuscript as Reference 33 (line 550).
Reviewer 2 Report
The manuscript describes a nice systematic study about xerogel nanocomposites which were obtained by tailoring the initial resorcinol-formaldehyde synthesis with Bi and Fe precursors.
Some parts of the Introduction are not well covered by references.
Line 141: … using A FEI Quanta 3D FEG … - capital A is not necessary.
Please, comment the stability of xerogel structures! Could it be destroyed easily, e.g. by ultrasound treatment?
What is the specific reason that electrochemical measurements were done only in the acidic region?
D and G signals, characteristic for to carbon structures should be marked in Fig. 1.II. as well.
Did you observe magnetic properties of Fe2O3 and Fe3O4? If yes, what is their role?
Is there any plausible reason that CXBiFe1.2 sample with the highest Fe concentration shows different type of isotherm (Fig. 1)?
Line 268-269: … (TBi = 271 °C K and TBi2O3=817 °C) … … (TFe = 1538 °C K and TFe3O4 ~ 1591-1597 °C), … °C or K???
Fig. 4: Sample GCE/CXBiFe0.12 cannot be seen – colored chart might help.
Fig. 7: (C) and (D) are not mentioned in the caption.
Author Response
The manuscript describes a nice systematic study about xerogel nanocomposites which were obtained by tailoring the initial resorcinol-formaldehyde synthesis with Bi and Fe precursors.
Some parts of the Introduction are not well covered by references.
In order to cover the Introduction, by references some citation were introduced in this part (see the newly added references [13-18]).
Line 141: … using A FEI Quanta 3D FEG … - capital A is not necessary.
This error was corrected, the article ‘A’ was re-written in lowercase (line 146).
Please, comment the stability of xerogel structures! Could it be destroyed easily, e.g. by ultrasound treatment?
After applying the drying and pyrolysis steps, the gels are found as monoliths with quite stable mechanical properties so that ultrasound treatments (in ethanol or water) are not expected to destroy the ultrastructure of the gels.
What is the specific reason that electrochemical measurements were done only in the acidic region?
The electrochemical analytical measurements were done in acetate buffer (pH 4.5) for heavy metal (Pb+2) detection, and phosphate buffer (pH 7) for H2O2 detection. The experimental conditions in this work were choose based on the previously reported results that proved that the higher electroanalytical parameters for Pb2+ detection were obtained in the presented experimental conditions [Fort, C.I.; Cotet, L.C.; Vulpoi, A.; Turdean, G.L.; Danciu, V.; Baia, L.; Popescu, I.C. Bismuth doped carbon xerogel nanocomposite incorporated in chitosan matrix for ultrasensitive voltammetric detection of Pb(II) and Cd(II). Sensors Actuators B Chem. 2015, 220, 712–719][ Rusu, M.M.; Fort, C.I.; Cotet, L.C.; Vulpoi, A.; Todea, M.; Turdean, G.L.; Danciu, V.; Popescu, I.C.; Baia, L. Insights into the morphological and structural particularities of highly sensitive porous bismuth-carbon nanocomposites based electrochemical sensors. Sensors Actuators B Chem. 2018, 268, 398–410].
D and G signals, characteristic for to carbon structures should be marked in Fig. 1.II. as well.
The D and G signals were marked in Figure 1.II as suggested. We also modified the caption text “Raman spectroscopy revealing the carbon specific vibrational region” into “Raman spectroscopy revealing the D and G carbon specific vibrational region” for accuracy.
Did you observe magnetic properties of Fe2O3 and Fe3O4? If yes, what is their role?
We observed a magnetic response for this type of xerogel nanocomposites, and we observed an increase in response to higher iron concentrations. Even though, the magnetic properties of the investigated materials cover an important issue, due to the occurrence of Fe2O3 and Fe3O4, and also due to Bi-O-Fe interactions, we restricted our investigations to the study of morphology, structure and electrochemical affinity towards heavy metals and hydrogen peroxide, mostly for simplicity and due to technical constraints.
We have observed, at higher Fe concentrations, that Fe2O3 in maghemite/hematite is more evidenced than Fe3O4 but not through their magnetic behavior but through surface analysis and Raman spectroscopy. A systematic investigation of the magnetic properties of such materials may be considered in future works.
Is there any plausible reason that CXBiFe1.2 sample with the highest Fe concentration shows different type of isotherm (Fig. 1)?
We have mentioned in the text several effects that can be met in such systems (lines 248-255): (1) role of Fe based particles as reinforcing agent in the pore structure, (2) generating micropore responses due to mesopore occupancy, (3) local increase in temperature around Fe sites. There was a lot of debate regarding this issue, one other hypothesis was that Fe influences the local acidity of the system during the wet-synthesis step when the initial pore network is formed and that these changes in the chemical environment may also lead to the changes observed with the increase in Fe content.
Line 268-269: … (TBi = 271 °C K and TBi2O3=817 °C) … … (TFe = 1538 °C K and TFe3O4 ~ 1591-1597 °C), … °C or K???
The errors pointed were corrected. Further on, we modified the sentence with “Having in mind that Bi and Bi2O3 have smaller bulk melting temperatures (TBi = 271 °C and TBi2O3= 817 °C) than Fe and iron oxides (TFe = 1538 °C and TFe2O3-Fe3O4 ~ 1567-1597 °C)” so that differences between the melting temperatures of the iron oxide phases should not be neglected. (lines 277-278)
Fig. 4: Sample GCE/CXBiFe0.12 cannot be seen – colored chart might help.
The referee suggestion was used to improve the quality of the Figure 4. The improvement was done in the text (line 315).
Fig. 7: (C) and (D) are not mentioned in the caption.
The correction was done in the text (line 387).